# Theoretically Revealing the Response of Intermolecular Vibration Energy Transfer and Decomposition Process of the DNTF System to Electric Fields Using Two-Dimensional Infrared Spectra

**DOI:** 10.3390/ijms24054352

**Published:** 2023-02-22

**Authors:** Haichao Ren, Linxiang Ji, Xianzhen Jia, Jun Tao, Ruipeng Liu, Dongqing Wei, Xiaofeng Wang, Guangfu Ji

**Affiliations:** 1Xi’an Modern Chemistry Research Institute, Xi’an 710065, China; 2Department of Physics and Engineering Physics, University of Saskatchewan, Saskatoon, SK S7N 5E2, Canada; 3College of Food Science and Engineering, Henan University of Technology, Zhengzhou 450001, China; 4College of Life Science and Biotechnology, Shanghai Jiao Tong University, Shanghai 200240, China; 5National Key Laboratory for Shock Wave and Detonation Physics Research, Institute of Fluid Physics, Chinese Academy of Engineering Physics, Mianyang 621000, China

**Keywords:** two-dimensional infrared spectra, electric filed, DNTF system, intermolecular vibration energy transfer, non-covalent interactions, decomposition process

## Abstract

The external electric field (E-field), which is an important stimulus, can change the decomposition mechanism and sensitivity of energetic materials. As a result, understanding the response of energetic materials to external E-fields is critical for their safe use. Motivated by recent experiments and theories, the two-dimensional infrared (2D IR) spectra of 3,4-bis (3-nitrofurazan-4-yl) furoxan (DNTF), which has a high energy, a low melting point, and comprehensive properties, were theoretically investigated. Cross-peaks were observed in 2D IR spectra under different E-fields, which demonstrated an intermolecular vibration energy transfer; the furazan ring vibration was found to play an important role in the analysis of vibration energy distribution and was extended over several DNTF molecules. Measurements of the non-covalent interactions, with the support of the 2D IR spectra, indicated that there were obvious non-covalent interactions among different DNTF molecules, which resulted from the conjugation of the furoxan ring and the furazan ring; the direction of the E-field also had a significant influence on the strength of the weak interactions. Furthermore, the calculation of the Laplacian bond order, which characterized the C-NO_2_ bonds as trigger bonds, predicted that the E-fields could change the thermal decomposition process of DNTF while the positive E-field facilitates the breakdown of the C-NO_2_ in DNTFⅣ molecules. Our work provides new insights into the relationship between the E-field and the intermolecular vibration energy transfer and decomposition mechanism of the DNTF system.

## 1. Introduction

The response of energetic materials to an accidental stimulus such as impact, shock, or heat, which can trigger an unintended detonation, is a persistent concern with regard to the safe usage of energetic materials. The electric field (E-field) is one of the most important external stimuli that has been employed as an electrically controlled approach for the ignition of energetic materials [1]. Several reports have indicated that the introduction of an external E-field into energetic materials can increase the amount of energy and consequently result in an increase in detonation pressure and velocity [2,3,4]. Tasker et al. reported that the application of a parallel E-field (0.02 V/nm) caused a significant reduction in the failure thickness of the explosive PBX-9502 [2]. In similar experiments, Lee studied the effects of the E-field on the initiation and growth of a reaction in an HMX-based, cast-cured explosive with the use of the modified gap test; modest effects were observed in the test [3,4]. In the actual experiment, the external E-field was imposed on energetic materials that were considered hazardous. In addition to the dangers that might result from experimental operations and equipment, such as strong E-field installations, high-energy molecules may be more unstable in the presence of an external E-field. Computational techniques have recently become popular for determining the behaviors of energetic materials in the presence of an external E-field [5,6,7,8,9,10,11,12]. Polizer and co-workers systematically studied the influences of an E-field (−1.5426~1.5426 V/nm) on the “trigger linkage” bonds, including C-NO_2_, N-NO_2,_ and O-NO_2_, in nitromethane and dimethylnitramine. The results indicated that these bonds could be strengthened along the E-field direction through the reinforcement of intrinsic molecular polarities [5,6,7]. Ren et al. theoretically established linear correlations between the strength of the E-field (−5.142~5.142 V/nm) and the structural parameters, including bond length and bond stretching frequency, and demonstrated that the E-field could trigger the rupture of the N-O bond at the initial stage of decomposition [8,9,10]. Liu et al. investigated the structural arrangement of nitromethane under an E-field ranging from 0 to 5 V/nm with the use of molecular dynamics (MD). They found that the critical E-field strength was between 2 and 3 V/nm, which could induce the transition of nitromethane molecules from a relatively disordered distribution to a solid-like, ordered, and compact arrangement with a large density [11]. Wood et al. studied the initial reactions and subsequent decomposition of α-HMX, which was excited via an E-field ranging from 0.56 V/nm to 3.56 V/nm at various frequencies through MD with the reactive potential ReaxFF. Their results revealed that the energy increased with an increase in the rate of the energy input and plateaued as the process became athermal for high loading rates [12]. However, there is still a lack of understanding regarding the role of the E-field in the intermolecular interactions of energetic materials.

As a promising high-density material, 3,4-bis (3-nitrofurazan-4-yl) furoxan (DNTF), with its high energy, low melting point, and excellent comprehensive properties, has become the focus of researchers studying energetic materials in recent years [13,14,15,16,17,18,19,20]. It was found that a coordinated oxygen atom exists in the furoxan ring and that the C-NO_2_ bond is the longest and most likely to be broken down upon heating or impact [13,14]. Nan et al. conducted experiments that determined the thermal decomposition properties of DNTF in the condensed phase. They employed differential scanning calorimetry and fast scanning Fourier transform infrared (FTIR) spectroscopy and found that in the thermal decomposition process, C-NO_2_ broke down first, followed by the breakdown of the C-C bond connecting the furazan and furoxan rings [15]. The fast thermolysis of DNTF at 0.1~0.4 Mpa was investigated using temperature-jump FTIR spectroscopy, and the results revealed that the two competitive reactions, C-NO_2_ homolysis (to form NO_2_) and isomerization (to form NO), could occur during the fast thermolysis of DNTF [16]. Theoretical and experimental works demonstrated that the loss of the nitro group (NO_2_) was an important step in the shock initiation of DNTF, and that the intermolecular non-covalent interactions contributed to the stability of DNTF.

The rate of energy transfer in the presence and absence of intermolecular non-covalent interactions is of interest to the greater explosive community, and it is important to understand the nature of these interactions in explosive materials [21,22,23,24,25,26,27,28]. Ostrander et al. reported two-dimensional infrared (2D IR) measurements of pentaerythritol tetranitrate (PETN) thin films that resolved vibrational modes and were coherently delocalized over at least 15–30 molecules through the measurement of the transition dipole strength [23]. Shi et al. investigated the structural dynamics and the vibrational energy transfer dynamics of hexahydro-1,3,5-trinitro-1,3,5-triazine (RDX) using 2D IR spectroscopy, and the results indicated that vibrational energy transfer was more efficient in RDX microcrystals than in the solvated form [24]. Using theoretical, quasi-static 2D IR spectra, we recently calculated the vibrational energy transfer rate between 2,4,6-trinitrotoluene (TNT) and 2,4,6,8,10,12-hexanitro-2,4,6,8,10,12-hexaazaisowurtzitane (CL-20) in the TNT/CL-20 cocrystal [25]. Considering the proposed importance of vibrational energy transfer in energy materials, 2D IR was especially well-suited for providing insight into the molecular structure and vibrational dynamics that may impact explosive design [26,27,28].

In this manuscript, we theoretically reveal the vibrational energy transfer process in the DNTF system using 2D IR spectra under different E-fields, with the use of the asymmetric NO_2_ stretching modes as the vibrational probes. Moreover, the stability of the trigger C-NO_2_ bonds and the non-covalent interactions were studied, respectively, using the Laplacian bond order (LBO) method [29] and the independent gradient model based on the Hirshfeld partition (IGMH) method [30]. This investigation serves as a theoretical reference for the efficient addition of an E-field to the DNTF system as well as a means to avoid a catastrophic explosion in an external E-field.

## 2. Results and Discussion

The theoretical and experimental geometries of DNTF in the absence of an E-field are presented in Table 1 [13,14], and the theoretical geometries of DNTF under different E-fields are given in Appendix A. It can be seen that the bond lengths of N7O_2_ are well consistent with the experimental values of 1.207 Å and 1.217 Å, while the bond lengths of N8O_2_ vary greatly and are slightly smaller than the experimental values. The theoretical bond lengths of the furoxan ring range from 1.301 Å to 1.477 Å, which agree with the experimental scope of 1.302~1.440 Å. Compared with the experimental values, the optimized furoxan rings are within the reasonable scope of 1.295 Å~1.433 Å. Moreover, the theoretical trigger bond lengths of C-N show excellent agreement with the experimental values of 1.459 Å and 1.442 Å. Finally, the average deviation between the values for the theoretical DNTF molecules and the experimental optimization is calculated, and the results indicate that DNTFⅠ has the smallest value at 0.0128 Å, while DNTFⅡ has the largest value at 0.0144 Å. 

Due to the fact that NO_2_ asymmetric stretching modes ranging from 1500 cm^−1^ to 1700 cm^−1^ are often used in the vibrational probes of energetic materials [21,22], details of the vibrational assignment of the NO_2_ asymmetric stretching vibration are presented in Figure 1 and Appendix A. The theoretical anharmonic IR peak at 1640.8 cm^−1^ is consistent with our experimental value of 1639.5 cm^−1^ and other reports [13,14,17,18], and it is attributed to the NO_2_ asymmetric and furazan ring stretching vibration in DNTFⅣ; however, there is an obvious difference in the relative strength. The theoretical anharmonic IR peak at 1612.0 cm^−1^ is ascribed to the NO_2_ asymmetric and furazan ring stretching vibration in DNTFⅡ, which is in agreement with the experimental value of 1612.5 cm^−1^ [13,14,17,18]. Moreover, there is a relatively strong experimental peak at 1586.4 cm^−1^ [13,14,17,18] that is confirmed by the theoretical IR peak at 1579.7 cm^−1^. This is contributed by the NO_2_ asymmetric and furazan ring stretching vibrations in DNTFⅡ and the NO_2_ asymmetric stretching vibrations in DNTFⅣ. However, some theoretical IR peaks have not been reported in experiments. For example, the theoretical peak at 1623.2 cm^−1^ is attributed to the NO_2_ asymmetric and furazan ring stretching vibration in DNTFⅢ and the furoxan ring stretching vibration in DNTFⅠ. The theoretical peak at 1640.8 cm^−1^ is ascribed to the NO_2_ asymmetric and furazan ring stretching vibrations in DNTFⅣ. Through the above comparisons of the structures and IR peaks, the conclusion is that the optimized DNTF structure can well reproduce the experimental one, despite some differences. 

The normalized quasi-static 2D IR spectra of NO_2_ asymmetric stretching modes under different E-fields are presented in Figure 2. The diagonal peaks of the 2D IR spectra reflect their corresponding infrared spectra, as shown in the top panel and bottom panel of Figure 2; their intensities are scaled with the transition dipole as μ [4,31,32]. As can be seen, a network of diagonal and off-diagonal peaks comprises a pair of positive (in blue) and negative (in yellow) signals, which are identified as the ground-state bleaching and simulated emission (*v* = 0→*v* = 1, where *v* is the vibration quantum number), and as the first excited-state absorption (*v* = 1→*v* = 2), respectively [33,34]. Due to relatively narrower absorption peaks, four main pairs of positive and negative signals are observed, and the positive signals usually contain several vibration modes from different molecules based on vibration energy distribution. However, the negative anharmonicity and small energy gap between the two NO_2_ modes in DNTF lead to the overlap of negative and positive peaks, while the E-field is 1.0284 V/nm (Figure 2d). The anharmonicity of the transition (ω = 1611.9 cm^−1^), which is attributed to the N7O_2_ asymmetric and the corresponding furazan ring stretching vibration in DNTFⅠ, is −19.8 cm^−1^, and the anharmonicity of the transition (ω = 1617.7 cm^−1^), which mainly comes from the NO_2_ asymmetric and furazan ring stretching vibration in DNTFⅡ, is 19.9 cm^−1^. Besides the diagonal peaks, cross-peaks are also seen in the upper left and lower right panels of Figure 2, which indicates that the vibrational energy transfer among the four eigenstates extends over multiple DNTF molecules. For example, when the E-field is 2.0568 V/nm (Figure 2e), there are four cross-peaks between the N8O_2_ asymmetric and the corresponding furazan ring stretching vibration in DNTFⅣ (ω = 1577.2 cm^−1^) and between the N7O_2_ asymmetric and the corresponding furazan ring stretching vibration in DNTFⅡ (ω = 1593.9 cm^−1^); this demonstrates that although DNTFⅣ and DNTFⅡ are spatially separated by DNTFⅢ, the vibration energy transfer between them still exists. Conversely, when the E-field is −1.0284 V/nm (Figure 2b), the cross-peaks between the N7O_2_ asymmetric stretching modes in DNTFⅡ (ω = 1626.6 cm^−1^) and DNTFⅢ (ω = 1656.4 cm^−1^) disappear, which indicates that the vibrational energy transfer process between them is not possible. The reasons are as follows: For linear spectroscopy in a coupled dimer, it is reasonable to consider the one-exciton Hamiltonian [33,34,35,36] (1)$$H=\left(\begin{array}{cc} \hbar \omega_{i} & \beta_{\mathrm{ij}}\\ \beta_{\mathrm{ij}} & \hbar \omega_{j} \end{array}\right)$$
where β and ω represent the coupling constant and the frequency between chromophore i and j, respectively. The exciton eigenvalues are given as follows
(2)$$E_{i,j}=\frac{\hbar \omega_{i}+\hbar \omega_{j}\mp \sqrt{4\beta_{\mathrm{ij}}^{2}+{\left(\hbar \omega_{i}-\hbar \omega_{j}\right)}^{2}}}{2}$$


The eigenstates of E_i_ and E_j_ are
(3)$$\left|\phi_{i}\rangle =\cos\alpha \right|10\rangle -\sin\alpha |01\rangle$$
(4)$$\left|\phi_{j}\rangle =\sin\alpha \right|10\rangle -\cos\alpha |01\rangle$$
where α is the mixing angle
(5)$$\tan2\alpha =\frac{2\beta_{\mathrm{ij}}}{\hbar \omega_{j}-\hbar \omega_{i}}$$


In the weak coupling regime, the coupling constants are smaller than the frequency splitting
(6)$$\left|\beta_{\mathrm{ij}}\right|\ll \left|\hbar \omega_{i}-\hbar \omega_{j}\right|$$

Their mixing angle α is small, and the exciton states will be localized on the individual sites. Furthermore, the two excitonic states closely resemble the uncoupled local modes
(7)$$E_{i,j}\approx \hbar \omega_{i,j}\mp \frac{2\beta_{\mathrm{ij}}^{2}}{\hbar \omega_{j}-\hbar \omega_{i}}$$

Then, the coupling constant β_ij_ can be obtained based on the perturbation theory
(8)$$\beta_{\mathrm{ij}}\propto \sqrt{\frac{\Delta_{\mathrm{ij}}}{\Delta_{i}}}\left|\omega_{i}-\omega_{j}\right|$$
where Δ_ij_ and Δ_i_ represent the off-diagonal anharmonicity and the diagonal anharmonicity, respectively.

Based on the Fermi golden rule, the energy transfer rate κ_ij_ is proportional to the square of the coupling constant β_ij_, that is
(9)$$\kappa_{\mathrm{ij}}\propto \beta_{\mathrm{ij}}^{2}$$

From Equations (8) and (9), when the off-diagonal anharmonicity Δ_ij_ is zero, that is, when the cross-peaks vanish, the energy transfer rate is also zero.

In addition, based on the vibration energy distribution and comparison of the 2D IR spectra, it can be concluded that the furazan ring stretching vibrations play an important role in intermolecular vibration energy transfer.

In the above discussions, it was established that the vibration energy transfer usually occurs among the vibration modes, including the furazan ring stretching vibration in DNTF. Considering the fact that intermolecular vibration energy transfer is the bridge of intermolecular interactions, we opted to use the IGMH method [30,37], where the Hirshfeld partitioning of the actual molecular electron density is used to derive the atomic densities in order to be able to analyze intermolecular non-covalent interactions in the DNTF system under different E-fields (Figure 3). To maximally reveal the interactions between the DNTF molecules, the isovalue was set to 0.01 a.u. in the maps. As expected, due to the conjugation of the furoxan and furazan rings, a series of thin and broad isosurfaces appear between adjacent DNTF molecules, which ideally exhibit π-π stacking interactions and can also explain the existence of cross-peaks in the 2D IR spectra. More specifically, there are mainly π-π stacking interactions between the nitro group and the furazan ring in DNTFⅠ and DNTFⅡ; between the furazan ring connecting with the N8O_2_ in DNTFⅡ and the furazan and furoxan rings in DNTFⅢ; and between the furoxan ring in DNTFⅢ and the furazan ring connecting with the N8O_2_ and the furoxan ring in DNTFⅣ. Moreover, a van deer Walls (vdW) interaction is observed between the nitro group in DNTFⅠ and that in DNTFⅢ. Interestingly, when the direction of the E-field is opposite to the *X*-axis (Figure 3a,b), the area of the green isosurface between DNTFⅢ and DNTFⅣ, especially between the furoxan ring in DNTFⅢ and the furazan ring connecting with the N8O_2_ in DNTFⅣ (Figure 3, top right), is larger than that when the E-field is along the *X*-axis (Figure 3d,e); this indicates that the direction of the E-field has a significant influence on the strength of weak interactions.

It is believed that a more sensitive energetic molecule generally includes a trigger bond with a smaller bond order calculated using quantum chemistry. Furthermore, the experimental results demonstrate that the thermal decomposition process in DNTF involves the C-NO_2_ bonds breaking down first [15,16,19,20]. Therefore, the stability of the C-NO_2_ bond as the trigger bond is investigated using the Laplacian bond order (LBO) [38]. The LBO, which corresponds to the Laplacian of electron density in the fuzzy overlap space, is more comprehensive in investigating the stability of covalent bonds and is defined between atoms A and B as follows [38]
(10)$$\nabla^{2}\rho (r)=\frac{\partial^{2}\rho (r)}{\partial x^{2}}+\frac{\partial^{2}\rho (r)}{\partial y^{2}}+\frac{\partial^{2}\rho (r)}{\partial z^{2}}$$
(11)$${\mathrm{LBO}}_{A,B}=-10\times \int_{\nabla^{2}\rho <0}w_{A}(r)w_{B}(r)\nabla^{2}\rho (r)dr$$
where ρ(r) denotes the electron density at point r, and w_A_(r) stands for an atomic weighting function and corresponds directly to the scope of the atomic space. The prefactor −10 is introduced to make the magnitude of the LBO consistent with one’s chemical intuition for typical covalent bonds. Generally, the value of the LBO is positively related to the stability of the covalent bonds. As the DNTF primitive cell contains four DNTF molecules [13,14], the LBO values of eight C-NO_2_ bonds are presented in Figure 4. Due to the orientation of the coordinated oxygen on the furoxan ring [13,14], the LBO values of C-N8O_2_ are usually larger than those of C-N7O_2_, which reveals that the C-N7O_2_ bonds tend to induce decomposition. More specifically, regardless of the E-field change, the breakdown of the C-N7O_2_ bond in DNTFⅠ is the first step in the thermal decomposition process. The LBO value of the C-N8O_2_ bond in DNTFⅡ is the largest and is almost unchanged, which may be attributed to the fact that N8O_2_ in DNTFⅡ has strong π-π stacking interactions with the furoxan and furazan rings in other molecules (Figure 3). When the E-field is at 1.0284 V/nm, the LBO value of the C-N7O_2_ bond in DNTFⅠ is the largest, demonstrating that the DNTF in this E-field is relatively more stable during the decomposition process. However, with an E-field change, the next steps in the decomposition process are competitive. For example, when the E-field is −2.0568 V/nm, the second and third steps involve the cleavage of the C-N7O_2_ bonds in DNTFⅡ and DNTFⅣ, respectively, which are the same as those in the absence of an E-field. Nevertheless, when the E-field is 2.0568 V/nm, the second and third steps in the decomposition process represent the cleavage of the C-N7O_2_ bonds in DNTFⅣ and that of the C-N8O_2_ bonds in DNTFⅢ, respectively.

The highest occupied molecular orbital (HOMO) and the lowest unoccupied molecular orbital (LUMO) play a significant role in chemical reactivity [39]. Particularly, their energy gap can reflect the chemical reactivity and optical polarizability of energy materials [40,41]. The distribution of the HOMO and LUMO of DNTF under E-fields, along with their energy gaps, is presented in Figure 5. It can be seen that the LUMO is mainly distributed on the N7O_2_ and the corresponding furazan ring in DNTFⅣ under negative E-fields, and that it is mainly distributed on the N8O_2_ and the corresponding furazan ring in DNTFⅡ in the absence of an E-field and under positive E-fields. The HOMO is contributed by the furoxan and furazan rings connecting with the N7O_2_ in DNTFⅡ in the absence of an E-field and under negative E-fields, but the HOMO is contributed by the furoxan and furazan rings connecting with the N7O_2_ in DNTFⅣ under positive fields. In addition, larger HOMO and LUMO energy gaps usually lead to lower chemical reactivity, which allows for molecules to have higher chemical stability [40,41]. It is also revealed that when the E-field is −2.0568 V/nm, the value of the energy gap is the smallest, indicating that the DNTF molecule is the most active; on the other hand, the value of the energy gap is the largest in the absence of an E-field, which suggests that the DNTF molecule has the highest chemical stability. Therefore, E-fields may be considered to have an important impact on the chemical stability and position of the HOMO and LUMO.

## 3. Methods and Materials

The molecule structure and atomic coordination of DNTF were taken from its crystal structure, determined by X-ray diffraction CCDC (Cambridge Crystallographic Data Center) number 665080 [13]. The primitive cell includes four DNTF molecules, wherein both nitro groups were determined to be twisted in relation to the adjacent furazan rings, and both furazan rings were twisted relative to the central furoxan ring (Figure 6 and Appendix A). Geometric optimization and vibration frequency calculations of DNTF were performed via density function theory (DFT) at the level of B3LYP (D3), with the 6-311G (d, p) basis set as implemented in the Gaussian 16 program and the in-house BDF program [33,34,35,38,42,43,44,45,46]. The static E-field along with the *X*-axis (Figure 6, red arrow) was in the range of −2.0568~2.0568 V/nm, which has been extensively investigated [5,6,7,8,9,10,11,12]. The positive values mean that the direction of the E-field is along the *X*-axis, while the negative values indicate that the direction of the E-field is opposite to the *X*-axis. The absence of negative frequencies revealed by harmonic vibrational analysis confirms that the structure is a true minima. Then, through the vibration energy distribution analysis (VEDA) code [36], the frequencies were designated as the normal vibration modes. Anharmonic vibrational analysis was carried out on the reduced subspace of the asymmetric NO_2_ stretching normal modes. The anharmonic treatment of the vibrations was performed using the generalized second-order vibrational perturbation theory (GVPT2) implemented in Gaussian 16 [34,47], including terms up to the third and fourth derivatives of the potential energy with respect to the normal mode coordinates.

The calculation of quasi-static 2D IR spectra has been described in detail elsewhere [25,31]. In summary, the ground-state, two-exciton, and combination band frequencies of the asymmetric NO_2_ stretching modes were used to construct the two-exciton Hamiltonian, where the anharmonicity was calculated with the use of anharmonic frequencies; the two-exciton Hamiltonian was subsequently diagonalized. The transition dipole moment vectors between the vibrational ground state and the one-exciton state for each of the normal modes were acquired from the DFT results [34]. Then, the harmonic approximation, that is, the corresponding ground state of one-exciton state transition dipole moment vector multiplied by √2 42, was used to calculate the transition dipole moment vector between the one-exciton and two-exciton states. Moreover, the waiting time T_w_ was set to 3.0 ps. In a given vibrational mode, the fourth power of the transition dipole moments involved is generally proportional to the 2D IR diagonal peak intensity [31,32].

## 4. Conclusions

In this work, with the use of DFT/B3LYP (D3)/6-311G (d, p) calculations, we performed a comprehensive investigation of the effects of E-fields on the explosive DNTF, which included the study of intermolecular vibration energy transfer processes and non-covalent interaction using 2D IR spectra and the IGMH method, respectively. When there was an absence of an E-field, the optimized structural parameters of the DNTF primitive cell containing four DNTF molecules were in good agreement with the available experimental results, and the anharmonic theoretical spectra of the NO_2_ asymmetric stretching well reproduced the experiment peaks. In combination with the vibration energy distribution, cross-peaks in the 2D IR spectra were observed, which revealed that intermolecular vibrational energy transfer is extended over the DNTF molecules under different E-fields and that the furazan ring vibrations play an important role in the intermolecular vibration transfer process. It could be seen on the isosurface of the weak interaction that obvious non-covalent interactions exist among different DNTF molecules due to the conjugation of the furoxan and furazan rings, which were confirmed by the cross-peaks in the 2D IR spectra. Furthermore, it was found that the direction of the E-field has a significant influence on the strength of the weak interactions. More importantly, the LBO value of the C-NO_2_ bonds acting as the trigger bonds under different E-fields indicated that the E-fields could change the thermal decomposition process of DNTF, with the non-covalent interactions significantly contributing to the relative sensitivity of energy materials and the positive E-field facilitating the breakdown of C-NO_2_ in DNTFⅣ molecules. When the E-field was 1.0284 V/nm, DNTF was relatively more stable during the decomposition process. Finally, measurements of the HOMO and LUMO energy gaps indicate that DNTF is activated by E-fields and that the strength of E-fields has a relatively significant influence on the chemical reactivity of DNTF.

## Figures and Tables

**Figure 1 ijms-24-04352-f001:**
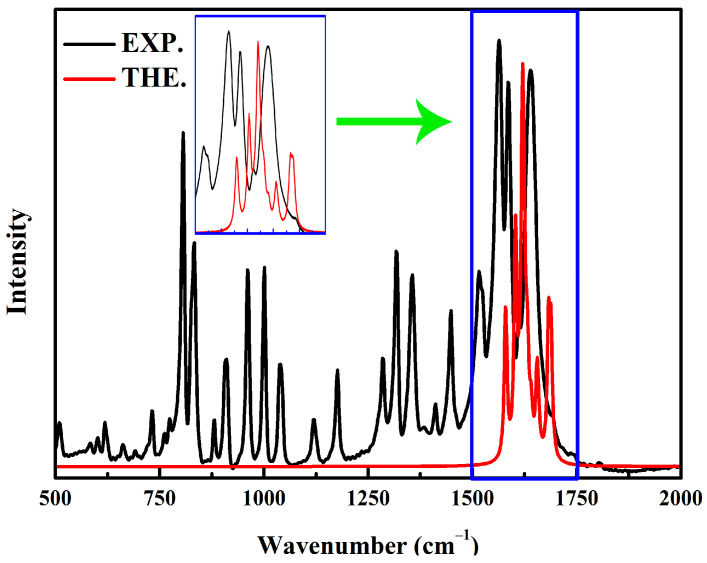
The experimental and theoretical IR spectra of DNTF molecules; the theoretical anharmonic IR spectra only include NO_2_ asymmetric stretching modes.

**Figure 2 ijms-24-04352-f002:**
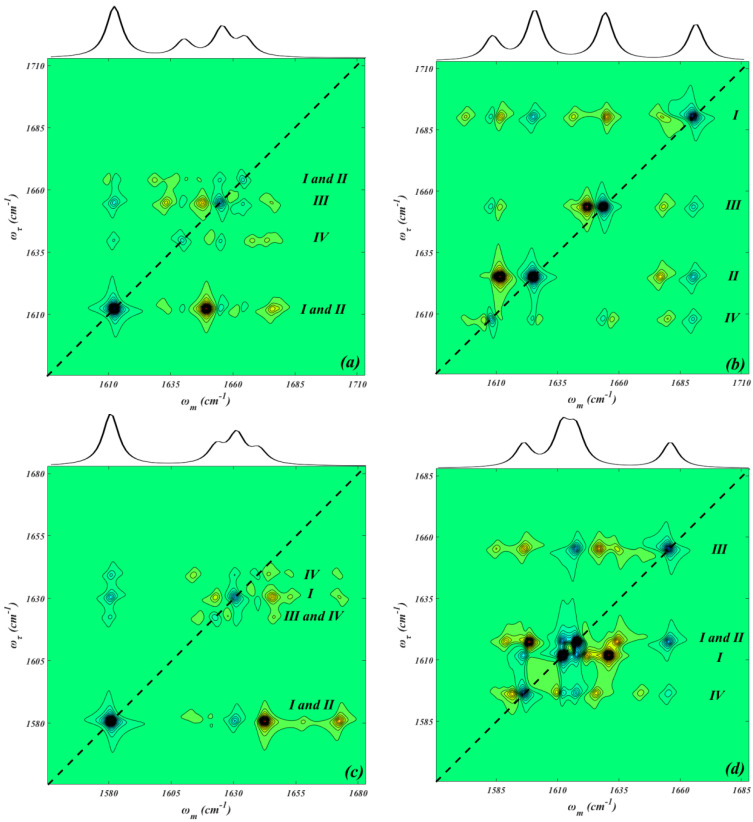
Normalized and simulated quasi-static 2D IR spectra and the corresponding IR spectra (top panel) of NO_2_ asymmetric stretching vibrations at different E-fields (V/nm). E-fields: (**a**) −2.0568; (**b**) −1.0284; (**c**) 0; (**d**) 1.0284; (**e**) 2.0568. Roman numbers represent the vibration modes in the corresponding DNTF molecules.

**Figure 3 ijms-24-04352-f003:**
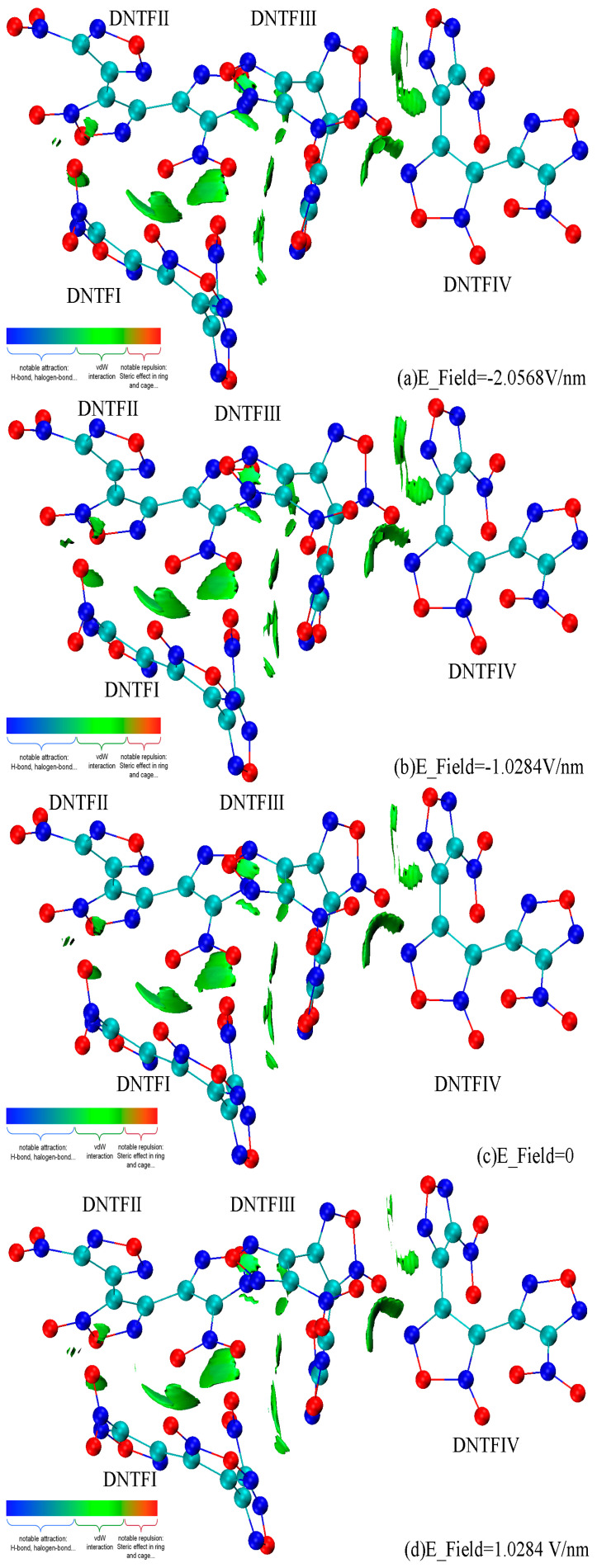
Intermolecular non-covalent interactions among DNTF molecules under different E-fields. The green isosurface represents the 0.01 a.u. region.

**Figure 4 ijms-24-04352-f004:**
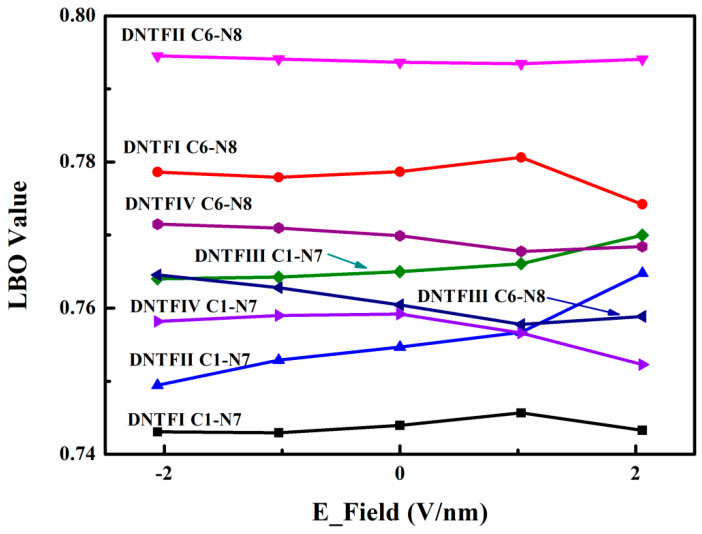
The LBO value of C-NO_2_ bonds in the DNTF system under different E-fields.

**Figure 5 ijms-24-04352-f005:**
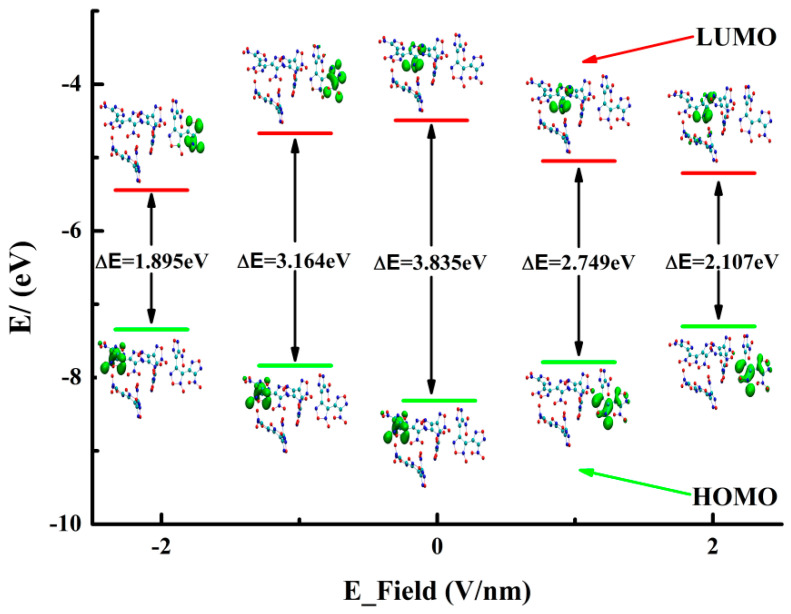
Variation trends for the ΔE_HOMO-LUMO_ of DNTF under different E-fields.

**Figure 6 ijms-24-04352-f006:**
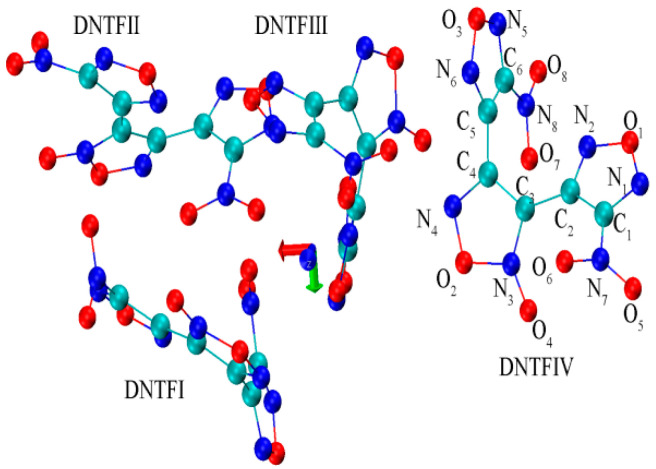
The optimized structure of the DNTF primitive cell includes four molecules, which are numbered using Roman numbers. The carbon (C), oxygen (O), and nitrogen (N) atoms are represented by blue-green, red, and blue-colored balls, respectively. Atoms in DNTFⅣ are labeled with elements and numbers, and the same rule is followed in other DNTF molecules.

**Table 1 ijms-24-04352-t001:** Theoretical and experimental bond lengths (Å) of DNTF in the absence of an E-field.

Bond	DNTFI	DNTFII	DNTFIII	DNTFIV	Experiment
N7-O6	1.213	1.211	1.218	1.211	1.207
N7-O5	1.217	1.221	1.218	1.221	1.217
C1-N7	1.467	1.464	1.459	1.464	1.459
C1-C2	1.433	1.429	1.426	1.429	1.421
C1-N1	1.294	1.297	1.300	1.297	1.289
N1-O1	1.360	1.357	1.350	1.356	1.372
N2-O1	1.364	1.367	1.367	1.368	1.375
N2-C2	1.311	1.311	1.308	1.311	1.303
C2-C3	1.441	1.441	1.444	1.442	1.445
C3-N3	1.337	1.337	1.329	1.336	1.336
N3-O4	1.201	1.198	1.207	1.201	1.212
C3-C4	1.428	1.424	1.422	1.421	1.405
N8-O7	1.234	1.216	1.224	1.224	1.255
N8-O8	1.204	1.220	1.210	1.211	1.227
C6-N8	1.454	1.447	1.463	1.461	1.442
C5-C6	1.426	1.424	1.425	1.424	1.415
C6-N6	1.297	1.295	1.297	1.297	1.296
O5-N6	1.353	1.351	1.357	1.356	1.366
O5-N5	1.374	1.375	1.374	1.374	1.379
N5-C5	1.305	1.305	1.305	1.304	1.304
C4-C5	1.464	1.465	1.463	1.465	1.472
C4-N4	1.303	1.304	1.302	1.301	1.302
O2-N4	1.349	1.348	1.354	1.353	1.372
O2-N3	1.465	1.477	1.469	1.470	1.440
Ave. Dev.	0.0128	0.0144	0.0134	0.0129	0

Note: $\mathrm{Ave}.\mathrm{Dev}.=\sqrt{\frac{\sum_{i}{\left({\mathrm{The}}_{i}-{\mathrm{Exp}}_{i}\right)}^{2}}{N}}$.

## Data Availability

The data presented in this study are available upon request from the corresponding authors.

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
