# Peer review of "Theoretically Revealing the Response of Intermolecular Vibration Energy Transfer and Decomposition Process of the DNTF System to Electric Fields Using Two-Dimensional Infrared Spectra"

_ijms, 2023, doi:10.3390/ijms24054352_

Round 1
Reviewer 1 Report
This paper reports the vibrational energy transfer process in 3, 4-bis (3-nitrofurazan-4-yl) furoxan (DNTF) employing 2D-IR spectra under different E-Field. This is a nice piece of work. There is one point to revise before the manuscript is accepted for publication in the International Journal of Molecular Sciences.
- Did the authors test other functionals and basis sets for geometry optimization and harmonic vibrational frequency calculations? In addition, the authors could consider including other dispersion corrections for DFT, as described in [PCCP, 2020,22, 8499-8512].
Author Response
Your approval of our work has given me lots of encouragement in future work. Thanks for your positive and constructive comments and suggestions. Geometry optimization and vibrational frequency calculations of DNTF were performed via DFT at the level of B3LYP (D3) with the 6-311G (d, p) basis set, and the results show an excellent agreement with the experimental data. Therefore, we didn’t test other functionals and basis sets for geometry optimization and anharmonic vibrational frequency calculations.
We have read the papers entitled “A generally applicable atomic-charge dependent London dispersion correction” and “Extension and evaluation of the D4 London-dispersion model for periodic systems”, and will apply the D4 London-dispersion into future works.

Reviewer 2 Report
In the submitted manuscript, the authors investigate by theoretical means (DFT level) the effect of external electric field on stability of an explosive material, DNTF. The focus is set on inter-molecular vibrational energy transfer within a four-molecule cluster corresponding to the material primitive crystal cell. The key finding is that the applied field can affect the energy-transfer pathways within the studied system, i.e., it alters inter-molecular vibrational couplings between the high-energy NO2 moieties. While I believe the undertaken topic can raise interest in the community, and the chosen methodology (in particular, the 2D-IR spectra simulations) provides valuable research tool, I am having troubles identifying the added value of the actual reported results. Eventually, I suggest a major revision before the manuscript could be accepted in IJMS, listing below my points of concern (not in order of importance).
1. English
The manuscript requires a detailed linguistic revision (punctuation, spelling, grammar, usage of field-related vocabulary, like Hamiltonian etc).
2. The structural optimization procedure
From the description given in the ‘Computational Methods and Theories’ section, it is still not clear to me how the structural relaxation of the DNTF geometry taken from the CCDC data base was performed: did the authors apply any boundary conditions on the four-molecule cluster cut from the crystal or used any other external point-charges potential to keep the shape of the cluster intact?
3. Unclear sentence on page 4
On the bottom on page 4, one reads ‘For the structure of DNTF with zero-hydrogen, the experimental IR peaks are less than 2000 cm-1.’ What does ‘DNTF with zero-hydrogen’ mean or refers to?
4. Figures 2 and 4 are partially unreadable
Even under large zoom, important parts of Figures 2 and 4 are far too small to read.
5. The effect of (large) external electric field on the molecular structure
Even though large external electric fields are employed in the study, their possible effect on the molecular crystal/cluster geometry is not discussed. At the same time, this factor could critically alter other studied properties like inter-molecular interactions, vibrational coupling etc.
6. The role of the applied electric field direction
In the ‘Conclusions’ section, the authors write: ‘Furthermore, the direction of the E-Field has a significant influence on the strength of the weak interactions.’ However, only one direction of the applied field is presented and discussed in the manuscript.
7. Figure 4 does not bring much and does not support strong impact of the field on inter-molecular interactions.
From the current form of Figure 4, illustrating inter-molecular interactions within the cluster, one cannot learn much more beyond the fact that… the applied electric field does really change much, which stays in some contradiction to the corresponding results discussion on page 8 (‘significant influence on the strength of the weak interactions’).
8. Discussion of data shown in Figure 5
The small, non-monotonic change of the DNTF-1: C1-N7 LBO value for the field value of ca. +1V/nm does not provide convincing support for the eventually formulated conclusion ‘When the E-Fields is 1.0284 V/nm, DNTF is relatively least sensitive.’ This is a minor change which could very well be an artifact of the method and, without additional justification for such field-stability effect, I don’t think so strong conclusion can be formulated.
Author Response
- English
The manuscript requires a detailed linguistic revision (punctuation, spelling, grammar, usage of field-related vocabulary, like Hamiltonian etc).
Our Response: Thanks for your positive suggestion. We rechecked the manuscript with the help of English editing of MDPI.
- The structural optimization procedure
From the description given in the ‘Computational Methods and Theories’ section, it is still not clear to me how the structural relaxation of the DNTF geometry taken from the CCDC data base was performed: did the authors apply any boundary conditions on the four-molecule cluster cut from the crystal or used any other external point-charges potential to keep the shape of the cluster intact?
Our Response: Many thanks for your constructive suggestion. To calculate the 2D IR spectra, the ground-state, two-exciton and combination band frequencies are used to construct the two-exciton Hamiltonian. The Gaussian 16 program has unique advantage in calculating the anharmonic frequencies in molecular cluster but shows an ordinary performance in periodic systems. The boundary conditions isn’t applied in the four-molecule cluster. With the geometry optimization of DNTF, the positions of external point-charges cannot be changed, so the external point-charges potential isn’t used.
- Unclear sentence on page 4
On the bottom on page 4, one reads ‘For the structure of DNTF with zero-hydrogen, the experimental IR peaks are less than 2000 cm-1.’ What does ‘DNTF with zero-hydrogen’ mean or refers to?
Our Response: Thanks for your helpful suggestion. We meant that the structure of DNTF does not include hydrogen atoms. After careful consideration, we deleted the sentence in the manuscript.
- Figures 2 and 4 are partially unreadable
Even under large zoom, important parts of Figures 2 and 4 are far too small to read.
Our Response: Thanks for your valuable suggestion. We replaced the Figure 2 and 4 in the manuscript.
Figure 2. The experimental and theoretical IR spectra of DNTF molecules, and the theoretical anharmonic IR spectra only includes NO2 asymmetric stretching modes.
Figure 4. Intermolecular non-covalent interactions among DNTF molecules under different E-Fields. The green isosurface represents the 0.01 a.u. region.
- The effect of (large) external electric field on the molecular structure
Even though large external electric fields are employed in the study, their possible effect on the molecular crystal/cluster geometry is not discussed. At the same time, this factor could critically alter other studied properties like inter-molecular interactions, vibrational coupling etc.
Our Response: Thanks for your constructive suggestion. The theoretical bond length of optimized DNTF under different E-Fields are added in supporting materials.
- The role of the applied electric field direction
In the ‘Conclusions’ section, the authors write: ‘Furthermore, the direction of the E-Field has a significant influence on the strength of the weak interactions.’ However, only one direction of the applied field is presented and discussed in the manuscript.
Our Response: We are appreciative of your valuable suggestions. The static E-Field along with the X-axis (Figure 1, red arrow) was in the range of -2.0568~ 2.0568 V/nm. The positive values mean that the direction of the E-Field is along the X-axis, while the negative values indicate that the direction of the E-Field is opposite to the X-axis. We added the sentence in the manuscript.
- Figure 4 does not bring much and does not support strong impact of the field on inter-molecular interactions.
From the current form of Figure 4, illustrating inter-molecular interactions within the cluster, one cannot learn much more beyond the fact that… the applied electric field does really change much, which stays in some contradiction to the corresponding results discussion on page 8 (‘significant influence on the strength of the weak interactions’).
Our Response: Thanks for your helpful suggestion. Based on the vibration energy distribution and comparison of 2D IR spectra, it is concluded that the furuzan ring stretching vibrations play an important role in intermolecular vibration energy transfer. The independent gradient model based on the Hirshfeld Partition (IGMH) method is chosen to support the conclusion from 2D IR spectra. From the Figure 4, it can be seen that a series of thin and broad isosurfaces (green) appear among the furuzan rings in adjacent DNTF molecules, which ideally exhibits π-π stacking interactions and also explain the existence of cross peaks in 2D IR spectra. Moreover, when the direction of E-Filed is opposite to the X-axis (Figure 4 a and Figure 4 b), the area of green isosurface between DNTFâ…¢ and DNTFâ…£ is larger than that while E-Filed is along the X-axis (Figure 4 d and Figure 4 e), indicating that the direction of the E-Fields has a significant influence on the strength of the weak interactions. We added the sentence in the manuscript.
- Discussion of data shown in Figure 5
The small, non-monotonic change of the DNTF-1: C1-N7 LBO value for the field value of ca. +1V/nm does not provide convincing support for the eventually formulated conclusion ‘When the E-Fields is 1.0284 V/nm, DNTF is relatively least sensitive.’ This is a minor change which could very well be an artifact of the method and, without additional justification for such field-stability effect, I don’t think so strong conclusion can be formulated.
Our Response: Thanks for your constructive suggestion. The Laplacian bond order (LBO) is obtained by integrating the negative part of Laplacian of electron density in bonding region and has been proved to be able to faithfully represent the actual bonding strength (“J. Phys. Chem. A, 2013, 117 (14), 3100”, “Carbon, 2020, 468” and “Cryst. Growth Des. 2023, 23 (1), 104”). We modified “When the E-Fields is 1.0284 V/nm, DNTF is relatively least sensitive” to “When the E-Fields is 1.0284 V/nm, DNTF is relatively more stable in the decomposition process” in the manuscript.

Reviewer 3 Report
I suggest the use of correction factors in the calculations of IR spectra, in order to improve the approximation to the experimental data.
Author Response
We are appreciative of your constructive suggestions. The theoretical IR spectra in Figure 2 is the anharmonic spectra of the asymmetric NO2 stretching modes, which includes the anharmonic effect. In our knowledge, there is no convincing the correction factors for the anharmonic IR spectra.

Round 2
Reviewer 2 Report
The authors have accomodated my requests to the level I think is adequate to publish the proposed manuscript.
Author Response
Thanks for your positive and constructive comments and suggestions.
